# Myeloid-Derived Suppressor Cells: Therapeutic Target for Gastrointestinal Cancers

**DOI:** 10.3390/ijms25052985

**Published:** 2024-03-04

**Authors:** Junaid Arshad, Amith Rao, Matthew L. Repp, Rohit Rao, Clinton Wu, Juanita L. Merchant

**Affiliations:** 1University of Arizona Cancer Center, GI Medical Oncology, Tucson, AZ 85724, USA; junaidarshad@arizona.edu; 2Banner University Medical Center—University of Arizona, Tucson, AZ 85719, USA; raoa@arizona.edu (A.R.);; 3College of Medicine, University of Arizona, Tucson, AZ 85719, USA; matthewrepp@arizona.edu; 4University Hospitals Cleveland Medical Center, Case Western Reserve School of Medicine, Cleveland, OH 44106, USA; rohit.rao@uhhospitals.org; 5Division of Gastroenterology and Hepatology, Department of Medicine, University of Arizona College of Medicine, Tucson, AZ 85724, USA

**Keywords:** MDSC, gastrointestinal cancer, tumor microenvironment, CCR5, IDO-1, PDE, CXCR2

## Abstract

Gastrointestinal cancers represent one of the more challenging cancers to treat. Current strategies to cure and control gastrointestinal (GI) cancers like surgery, radiation, chemotherapy, and immunotherapy have met with limited success, and research has turned towards further characterizing the tumor microenvironment to develop novel therapeutics. Myeloid-derived suppressor cells (MDSCs) have emerged as crucial drivers of pathogenesis and progression within the tumor microenvironment in GI malignancies. Many MDSCs clinical targets have been defined in preclinical models, that potentially play an integral role in blocking recruitment and expansion, promoting MDSC differentiation into mature myeloid cells, depleting existing MDSCs, altering MDSC metabolic pathways, and directly inhibiting MDSC function. This review article analyzes the role of MDSCs in GI cancers as viable therapeutic targets for gastrointestinal malignancies and reviews the existing clinical trial landscape of recently completed and ongoing clinical studies testing novel therapeutics in GI cancers.

## 1. Introduction

Malignancies of the gastrointestinal (GI) tract represent some of the most difficult cancers to manage in clinical practice. Despite combinations of surgery, radiation, chemotherapy, and sometimes immunotherapy, the 5-year survival in GI malignancies (all stages combined) ranges from a high of 65% in colorectal cancer (CRC) to 12% in pancreatic cancer [1]. As targeting tumor cells with standard chemoradiation and surgery has limitations in ensuring long-term tumor control, cancer research has turned to characterization of the tumor microenvironment (TME) to ensure better outcomes in anticancer therapy. The tumor microenvironment plays a crucial role in the pathophysiology of GI cancers, influencing tumor growth, invasion, and response to therapy. Myeloid-derived suppressor cells (MDSCs) are a major component of the TME and play a crucial role in immune suppression and pathogenesis of GI cancers [1,2]. This narrative review will explore the current landscape of MDSCs involvement in GI cancers and their potential role as a therapeutic target in light of ongoing clinical trials.

## 2. Development and Classification of MDSCs

Myeloid-derived suppressor cells arise from disruption of normal myelopoiesis. Myeloid progenitors in the bone marrow differentiate into immature myeloid cells (IMCs) which further differentiate to form mature macrophages, granulocytes, or dendritic cells [3]. However, in cancer or chronic inflammation, outside stimuli lead to an increase in Janus kinase/signal transducers and activators of transcription 3 (JAK/STAT3) signaling in IMCs, blocking their normal differentiation, and leading them to acquire immunosuppressive activity [3]. These new cells are termed as myeloid-derived suppressor cells (MDSCs). After differentiation, suppressive activity of MDSCs can be further activated by toll-like receptor (TLR) and cytokine signaling [3]. MDSCs are divided into subsets based on their immature myeloid progenitor of origin. Polymorphonuclear (PMN)-MDSCs or granulocytic (G)-MDSCs arise from immature myeloid progenitors normally destined to become granulocytes [4]. In humans, PMN-MDSCs are defined by cells expressing cell surface markers CD11b+ CD15+ CD14- or CD11b+ CD14-CD66b+ [4]. The other major subset of MDSCs that arise from immature myeloid progenitors and are destined to become macrophages are designated as monocytic MDSCs (M-MDSCs). In humans, M-MDSCs are defined as expression of cell markers CD14+ CD33+ HLA-DR^LOW^ [5]. A third population of MDSCs in humans is defined as lineage negative (CD3/CD14/CD15/CD19/CD56 negative) HLA-DR negative CD33+, corresponding to early MDSC progenitor cells (eMDSC) [4]. Emerging evidence underscores the importance of MDSCs in the tumorigenesis of GI malignancies and resistance to therapies.

## 3. MDSCs Drive Transformation of Premalignant States in GI Malignancies

Recruitment of MDSCs is an early step in the tumorigenesis of multiple GI malignancies [6,7,8]. In gastric cancer, PMN-MDSCs that express *SLFN-4* have been identified in early gastric metaplasia following *Helicobacter* infection [6]. Early expression of interleukin -1B (IL-1B), a cytokine that recruits MDSCs, in the stomach of transgenic mice drives inflammation and early malignant transformation [9]. In esophageal squamous carcinoma, infiltration of MDSCs is required for tumorigenesis induced by a high-fat diet, and attenuation of CD8-driven immunosurveillance by PMN-MDSCs helps drive metaplasia in a mouse model of Barrett’s esophagus [10,11]. In colorectal cancer (CRC), expression of chemokine (C-C motif) ligand 2 (CCL2) drives migration of PMN-MDSCs into the tumor, suppressing immune responses and enhancing tumor progression [12]. In addition, CXCR2 (C-X-C chemokine receptor 2)-mediated infiltration of PMN-MDSCs into the CRC microenvironment is required for tumorigenesis induced by azoxymethane/dextran sodium sulfate (AOM/DSS) colitis [7]. Finally, elevated levels of PMN-MDSC and M-MDSC were also seen in the blood of patients with intraductal papillary mucinous neoplasm (IPMN), a premalignant pancreatic cancer lesion [8]. Early infiltration of MDSCs into premalignant lesions may suppress early antitumor immune responses that clear these lesions before they fully transform. Targeting MDSC infiltration in patients with these lesions may be a strategy that can help reduce disease progression.

## 4. MDSCs Drive Metastasis in GI Malignancies

Metastatic dissemination is an important driver of morbidity and mortality in GI malignancies. In CRC and pancreatic cancer, MDSCs play an important role in enabling this process. PMN-MDSC depletion inhibits progression of CRC peritoneal disease [13], while CXCR2 inhibitor can reduce the rate of metastasis by reducing the MDSC infiltration [14,15]. Finally, CXCR2+ MDSCs in the premetastatic niche promote CRC tumor cell survival [16], and M-MDSCs promote the growth of CRC micro-metastases to macro-metastases [17]. In pancreatic cancer, increased M-MDSCs are associated with increased invasion and epithelial–mesenchymal transition, leading to metastatic spread [18]. In GI malignancies, MDSCs help enable invasion, survival, and outgrowth of malignant tumor cells at the metastatic site, leading to immune evasion. Targeting MDSCs could therefore be an important strategy to inhibit tumor metastasis.

## 5. MDSCs Drive Recurrence in Response to Conventional Local and Systemic Therapies in GI Malignancies

GI malignancies are treated with a combination of locally directed therapies such as surgery, radiation, trans-arterial radioembolization (TARE), ablation, and systemic chemotherapies. MDSCs have been shown to play an important role in regrowth of colorectal, hepatocellular carcinoma (HCC), and pancreatic cancer following local therapy [19]. Palliative radiofrequency ablation in a model of liver metastatic CRC and surgical trauma drives recruitment of immunosuppressive MDSCs, enhancing tumor growth [19,20]. Radiation therapy promotes infiltration of M-MDSCs into the CRC TME and boosts their suppressive function [21,22]. In pancreatic cancer, radiation therapy drives recruitment of PMN-MDSCs and suppresses radiation responses [23]. MDSC infiltration is also associated with recurrence after local therapies in hepatocellular carcinoma. For example, the frequency of MDSCs after hepatic artery infusion therapy, yttrium-90 TARE, or radiotherapy is inversely linked with prognosis [24,25,26]. Following surgery, cytokine alpha interferon (IFNa) secretion drives recruitment of M-MDSCs, suppressing the activity of CD8-T-cells to drive recurrence [27]. Furthermore, M-MDSCs are increased in patients who have recurrent cancer after liver transplantation in a C-X-C motif chemokine ligand 10 (CXCL-10) dependent manner and drive HCC recurrence [28].

MDSCs drive resistance to systemic therapies in GI malignancies by decreasing antitumor immune responses as well as through direct modulation of tumor cells. M-MDSC co-cultured with esophageal squamous cancer cells (ESCCs) show decreased apoptosis in response to cisplatin therapy, possibly through induction of a cancer stem cell state [29]. Complete response to neoadjuvant chemotherapy in advanced esophageal adenocarcinoma is associated with decreased M-MDSC signatures [30], and the presence of CD33+ MDSC in gastric cancers is associated with increased risk of recurrence [31]. In colorectal cancer, MDSCs promote resistance to oxaliplatin therapy through facilitating epithelial to mesenchymal transition, which is associated with increased expression of drug efflux transporters and invasiveness [32]. Higher levels of MDSCs were associated with a worse prognosis in CRC patients treated with a combination of chemotherapy drugs including FOLFOX (5-Florouracil, Oxaliplatin) plus bevacizumab [33], and poor response to neoadjuvant chemotherapy in locally advanced rectal cancer [34]. Infiltration of PMN-MDSCs may be driven by dying CRC tumor cells, which release the ubiquitous translationally controlled tumor protein (TCTP) to promote MDSC recruitment [35]. Among conventional chemotherapy regimens for CRC, FOLFOX decreases immunosuppression, while FOLFIRI (5-Florouracil, Irinotecan) enhances it by decreasing MDSC apoptosis [36]. In HCC, MDSC infiltration suppresses immune responses after chemotherapy and increases posttherapy growth factor signaling. Oxaliplatin-resistant HCC lines showed increased recruitment of PMN-MDSCs in vivo through chemokine (C-C motif) ligand 5 (CCL-5) secretion [37]. Chemotherapy with 5-Florouracil increases infiltration of PMN-MDSCs, decreases T-cell infiltration, and drives resistance to immune checkpoint blockade [38]. Finally, PMN-MDSCs can promote resistance to sorafenib through fibroblast growth factor 1 (FGF1) secretion [39]. In summary, conventional chemotherapy, radiation, and surgery drive MDSC infiltration in GI malignancies, driving the switch from regression to recurrence by suppressing T-cell responses, inducing stem-like drug-resistant states and increased growth factor signaling.

## 6. MDSCs Drive Resistance to Immunotherapies in GI Malignancies

In current clinical practice, immunotherapies block the interaction of program death protein (PD1) with its ligand (PD-L1) or cytotoxic T-lymphocyte associated protein (CTLA4) axis to reverse T-cell exhaustion, increase T-cell proliferation, and enhance the antitumor immune response. MDSCs can antagonize the process through depletion of extracellular arginine, increase in immunosuppressive metabolites like adenosine and kyeurnine, reactive oxygen species, and direct expression of immunosuppressive ligands like PD-L1 [40]. MDSCs play an important role in driving resistance to immunotherapy into multiple GI malignancies. In a mouse model of gastric cancer, inhibition of Cysteine-Cysteine Motif chemokine receptor 5 (CCR5) mediated PMN-MDSC accumulation enhances efficacy of anti-PD-1 antibody therapy potentially through reversal of T-cell exhaustion [41,42]. Similar effects can be seen by depleting MDSCs using FOLFOX in gastric cancer [43]. Furthermore, in the KEYNOTE-061 trial for advanced gastric cancer, an M-MDSC signature was associated with a decreased overall response rate, progression-free survival (PFS), and overall survival (OS) in patients treated with pembrolizumab [44,45].

MDSCs also drive immunotherapy resistance in liver tumors. In intrahepatic cholangiocarcinoma (ICC), PMN-MDSCs accumulate in the TME and promote ICC progression; blockade of PMN-MDSC infiltration enhances the efficacy of anti-PD1 therapy [46]. Furthermore, single-cell RNA sequencing (scRNA-seq) identified a subset of apolipoprotein E gene (ApoE+) PMN-MDSCs that act as an important source of PD-L1 in the tumor microenvironment of ICC and helps suppress response to immune checkpoint blockade [47]. HCC arising from nonalcoholic steatohepatitis (NASH) is intrinsically resistant to immune checkpoint blockade compared to other etiologies of HCC. Increased infiltration of MDSCs may underlie this resistance. Spatial transcriptomics comparing NASH-HCC, hepatitis B virus (HBV)-HCC, hepatitis C virus (HCV)-HCC, and healthy controls show an increased proportion of PMN-MDSC, and M-MDSC in NASH-HCC compared to other etiologies, and an increased proportion in HCC in general compared to healthy controls [48]. Spatial profiling also showed that infiltrating T-cells in NASH-HCC were close in location to PD-L1+ PMN and M-MDSC, suggesting that these cells were important for immunosuppression [49]. In addition, secondary resistance to immune checkpoint blockade in human and mouse HCC is also driven by peroxisome proliferator-activated receptor gamma (PPARγ)-induced MDSC expansion [49]. 

Finally, MDSCs also are associated with immune checkpoint blockade resistance in colorectal and pancreatic cancer. Infiltration of IL-1B-positive MDSCs identified from scRNA-seq is associated with resistance to checkpoint blockade in microsatellite instability high (MSI-H) metastatic colorectal cancer patient [50]. In the COMBAT trial, treatment of pancreatic tumors with a C-X-C motif chemokine receptor 4 (CXCR4) antagonist decreased infiltration of MDSCs and was synergistic with immune checkpoint blockade in pancreatic cancer [51]. Combination therapy using histone deacetylase (HDAC) inhibitor entinostat with immune checkpoint blockade decreases M-MDSC infiltration and decreases the suppressive activity of PMN-MDSC in mouse pancreatic ductal adenocarcinoma (PDAC) models, leading to improved survival [45]. In multiple GI malignancies, infiltration of MDSCs suppresses responses to immunotherapy.

## 7. Targeting MDSCs in GI Malignancies

This review offers a comprehensive snapshot of available targets for MDSCs in GI malignancies employing both direct and indirect approaches. Classification of treatment strategies include (1) blocking MDSC recruitment and expansion, (2) promotion of MDSC differentiation, (3) MDSC depletion, and (4) alteration of MDSC metabolic pathways, and (5) inhibition of MDSC function [52].

Given the importance of MDSCs in driving progression and recurrence in GI malignancies, it is important to identify ways to target them. One method of MDSC targeting is to inhibit the pathways that govern entry of MDSCs in the immune environment. Over time, blocking recruitment of MDSCs will alter the balance of anti- and proinflammatory cells in the tumor microenvironment. The CCR5, CXCR1/2, and Phosphoinositide 3-Kinase (PI3-kinase) pathways all are being explored as mechanisms to target MDSC recruitment (Figure 1).

### 7.1. Blocking Migration (Recruitment/Expansion) of PMN-MDSCs—Cysteine-Cysteine Motif Chemokine Receptor 5 (CCR5) Signaling

CCR5, a G-protein-coupled receptor, is predominantly expressed on the surface of immune cells and interacts with various chemokine ligands, notably CCL3, CCL4, and CCL5 [53]. Signaling through CCR5 was recently found to be required for mobilization of PMN-MDSCs from the bone marrow [54]. CCR5 antagonism has shown to be synergistic with immunotherapy in gastric cancer [42]. Based on these findings, CCR5 is being explored as a target for inhibiting migration of PMN-MDSCs into GI cancers. Maraviroc is an FDA-approved inhibitor of CCR5 for treatment of HIV and is safe and well tolerated. In the MARACON-01 Phase I trial (Table 1), maraviroc (NCT01736813) was tested in 12 patients having colorectal cancer with liver metastases who had received an average of 4.3 lines of previous therapy [55]. Initial data were promising, with a median survival of 5.06 months, and three out of five partial responses in a subset of patients who decided to return to chemotherapy with maraviroc [55]. The same group followed this trial up with the PICASSO Phase I trial, in which the activity of combined CCR5 inhibition through maraviroc and anti-PD1 therapy (pembrolizumab) was explored in mismatch-repair-proficient metastatic colorectal cancer [56]. However, while this combination was well tolerated in the study, the majority of patients (18/19) showed progressive disease with a median progression-free survival (PFS) of 2.10 months [56]. Table 1 references a similar Phase II trial combining CCR5 inhibitor vicriviroc and pembrolizumab which showed a 5% objective response rate (ORR)and a median PFS of 2.1 months (NCT03631407) [57]. A Phase I single arm study examining the safety, tolerability, and feasibility of maraviroc with ipilimumab, and nivolumab in mCRC and pancreatic cancer is ongoing (Table 1) with results pending (NCT04721301) [58].

In addition to maraviroc and its analogues, other CCR5 inhibitors are being tested in trials. Table 1 references a phase Ib trial using the CCR5 antagonist OB-002 (NCT05940844) that is currently underway to assess the safety, tolerability, and pharmacokinetics in mCRC, pancreatic, and gastric cancer patients [59]. In borderline resectable or locally advanced pancreatic cancer, a trial of CCR2/5 inhibitor BMS-813160 (Table 1) in combination with gemcitabine, nivolumab, and nab-paclitaxel showed an overall response rate of 26.1%, with 32% of tumors becoming resectable after treatment (NCT03496662) [60]. Other ongoing studies include a Phase I/II trial of combination immunotherapy including nivolumab and a dual CCR2/5 antagonist BMS-813160 with a GVAX cancer vaccine (Table 1) in locally advanced pancreatic adenocarcinomas (NCT03767582) [61]. BMS-813160 is also being explored in a Phase II trial in hepatocellular carcinoma (NCT04123379) [62]. 

### 7.2. C-X-C Chemokine Receptor Type 1/2 (CXCR1/2)

The CXCLs–CXCR1/2 axis (Figure 1) plays a pivotal role in regulating the migration of neutrophils to sites of inflammation, including PMN-MDSCs to the TME [63]. Inhibiting CXCR2 not only slows tumorigenesis but prevents metastasis and sensitizes tumors to immune checkpoint blockade [63]. The STOPTRAFFIC-1 (NCT04599140) trial is using this rationale to investigate SX-682, a CXCR1/2 inhibitor, alone and in combination with nivolumab in patients with refractory rat sarcoma (*RAS*) gene mutated, microsatellite stable (MSS) metastatic CRC [64]. Similarly, a Phase II trial is using SX-682 with anti-PD1 antibody tislelizumab in the neoadjuvant setting for resectable pancreatic cancer after ≥1 prior chemotherapy regimen [65]. The exploratory outcomes include assessing changes in intra-tumoral granzyme B+ CD137+ T-cell density before and after neoadjuvant treatment. Simultaneously, SX-682 in combination with nivolumab is also being tested for metastatic pancreatic adenocarcinoma (Table 1) with 16 weeks stable disease on a first-line regimen (NCT04477343) [66]. A Phase I/II trial of SX-682 in combination with bintrafusp alfa which targets PD-L1, transforming growth factor beta (TGF-β), and CV301, a poxviral vaccine designed against carcinoembryonic antigen (CEA) and mucin 1 (MUC1) tumor antigens in advanced refractory pancreaticobiliary cancer showed durable biochemical responses and disease control [67]. Forty-six percent (n = 11) of patients who received the triplet experienced grade ≥3 bleeding [67].

AZD5069 is another CXCR1/2 inhibitor that has been tested in GI malignancies. A combination of durvalumab and AZD5069 was also tested in metastatic pancreatic cancer (EUCTR2015-003639-37-GB) [68]. In this trial, 4/12 patients suffered from dose-limiting toxicities, with 3 having to discontinue the study drug. Around 18/20 patients had an adverse event (AE) of common terminology criteria of adverse events (CTCAE) grade 3 or greater [68]. The ORR was 5.6%, median PFS was 1.6 months, and median OS was 2.8 months. A Phase I/II trial is looking at combination of CXCR1/2 inhibitor AZD5069 with anti-PDL1 inhibitor durvalumab in patients with unresectable HCC who are treatment-naive or have progressed on no more than one line of therapy (ISRCTN12669009) [69].

### 7.3. Phosphoinositide 3-Kinase (PI3K)

The PI3K-AKT signaling pathway is involved in induction of immunosuppressive molecules, recruitment of MDSCs into the TME, and chemokine-mediated chemotaxis (Figure 1). Notably, PI3-kinase signaling is a downstream effector signaling pathway for many upstream receptors that play important roles in driving MDSC migration and survival, such as CXCR2 [66]. PI3-kinase has four isoforms, p110α (PI3Kα), p110B (PI3KB), p110γ (PI3Kγ), and p110delta (PI3Kdelta) [70]. The first two isoforms are expressed ubiquitously, in tumor tissue, but PI3Kγ is selectively expressed in myeloid cells. Inhibition of PI3Kγ signaling has been shown to decrease recruitment of MDSCs [68]. Duvelisib is an FDA-approved PI3K γ/delta inhibitor for chronic lymphocytic leukemia (CLL) that has been shown to decrease MDSC induction of Arginase 1 (Arg1) and nitric oxide synthase 2 (NOS2) transcripts and enhance immunotherapy in preclinical models when given at low doses [71]. A Phase I/II trial by a Chinese group is exploring the combination of duvelisib with anti-PD-1 immunotherapy in advanced solid tumors (Table 1), with results still pending (NCT05508659) [72]. As PI3K-delta is expressed by T-cells as well, development is also underway for selective PI3Kγ inhibitors. A Phase I/II trial is studying the selective PI3Kγ inhibitor ZX-4081 in patients with advanced solid tumors (Table 1), including CRC and HCC (NCT05118841) [73].

### 7.4. Inducible Nitric Oxide Synthase (iNOS) and Vascular Endothelial Growth Factor (VEGF)

Tumor angiogenesis drives the progression of tumors, influencing their growth, invasion, and metastatic potential. The preeminent drivers of this process are the growth factors belonging to the VEGF family. While primarily recognized for their proangiogenic effects, VEGF also plays a crucial role in the recruitment of MDSCs to the TME and impedes dendritic cell differentiation, resulting in immunosuppression [74]. In a melanoma murine model, Jayaraman et al. demonstrated that iNOS overexpression causes VEGF secretion from tumor cells, which causes recruitment and activation of MDSCs [75]. The immunosuppressive activity of tumor-infiltrating MDSCs was reversed when mice were treated with L-NIL, a selective iNOS inhibitor, demonstrating an iNOS-dependent VEGF secretory pathway necessary for MDSC recruitment [75].

A Phase II trial in mCRC evaluated the activity and safety of anakinra, an inhibitor targeting both interleukin 1 (IL-1) α and β, along with 5-fluorouracil (5FU) and bevacizumab, a monoclonal antibody (mAb) targeting VEGF-A (NCT02090101) [76]. The rationale lies in 5-FU causing depletion of MDSCs, leading to activation of caspase-1 and IL-1β. This process polarizes CD4-T-cells into interleukin 17 (IL-17) secreting T-helper 17 cells (Th17), ultimately promoting VEGF production and angiogenesis in the TME. The trial showed a satisfactory response rate and an acceptable toxicity pattern [77].

### 7.5. Additional Approaches: Modulation of the Microbiome

The gut microbiome also may drive recruitment of MDSCs in GI malignancies [9]. Disruption of microbiome has long been known to correlate with increased susceptibility to infections and increased incidence of autoimmunity [9]. In GI malignancies, microbial dysbiosis can promote recruitment of MDSCs and tumor growth. Microbial infiltration into the liver from gut disruption drives recruitment of MDSCs and progression of ICC [9]. In mouse models of primary sclerosing cholangitis (PSC) or DSS colitis, inflammation damages the gut barrier, causing translocation of Gram-negative commensals into the liver [10]. This leads to TLR 4 and CXCR2 mediated recruitment of PMN-MDSCs, and accelerated ICC development [10]. Notably, PMN-MDSC populations were also increased in patients with PSC and active ulcerative colitis, compared to those with inactive ulcerative colitis or without it [10]. Similarly, in CRC, the presence of specific gut anaerobes drives MDSC infiltration into both metastases [11] and primary tumors [11]. In addition to bacteria affecting MDSC numbers, fungal dysbiosis can also enhance the suppressive function of infiltrating MDSCs in CRC through metabolic reprogramming [10,12]. Finally, the tumor microbiome has been shown to regulate MDSCs in pancreatic cancer. Transplant of bacteria from the gut of KRAS, p53, cyclic recombinase (KPC) mutated mice into antibiotic-treated mice of a slowly progressive KRAS, cyclic recombinase (KC) pancreatic cancer model accelerates tumor growth, but microbial ablation leads to decreased infiltration of MDSCs and reduces disease progression [7]. Cancer-associated fibroblasts (CAFs) are involved in recruitment of PMN-MDSCs into TME in gastric cancer, and PMN/M-MDSC infiltration in cholangiocarcinoma [8,13]. As antibiotics are drugs with a long record of clinical experience, the prospect of modulating the microbiome with specific antibiotic treatments combined with immunotherapy could be an attractive alternative to rapidly translate MDSC targeting to the clinic. 

A small Phase II trial has been performed testing the combination of tadalafil, oral vancomycin, and nivolumab in HCC or patients with liver metastases (NCT03785210) [78]. However, results only showed mostly progressive disease, with one patient in each treatment group showing stable disease [78]. Of note, while oral vancomycin has a limited spectrum of activity limited to Gram-positive, it is possible that the explanation for the results is a lack of coverage of other anaerobes. More research will have to be carried out before antibiotic treatment can be added to current regimens. 

## 8. Promotion of MDSC Maturation

In cancer or chronic inflammation, external stimuli lead to an increase in JAK/STAT3 and C/EBPβ signaling in IMCs, blocking their normal differentiation, and leading them to acquire immunosuppressive activity. There is also a decrease in transcriptional activity of the pro-differentiation transcription factor CCAAT/enhancer-binding protein alpha (C/EBPα). One way to suppress MDSC generation is by bypassing this block in differentiation in immature myeloid cells.

### 8.1. JAK2/STAT3 Pathway Inhibition

JAK2/STAT3 pathway inhibition affects MDSCs by inducing MDSC apoptosis and by promoting MDSC differentiation to mature myeloid cells (Figure 1) [79]. The MODULATE trial sought to modify the TME to enhance the anti-PD1 response in patients with MSS mCRC by randomizing 90 patients evenly into two arms [80]. The first arm received nivolumab in combination with BNC105, a vascular-disrupting agent, while the second arm received nivolumab with napabucasin, a STAT3 inhibitor [80]. Alterations in the TME were monitored through repeated baseline tumor biopsies at 6 and 12 weeks. The study showed an ORR 5% in both treatment groups [80]. The results proved the rejection of the null hypothesis set at 2%; however, the desired ORR of 15% was not achieved (NCT03647839) [80]. In a Phase Ib/II study, 92% of patients (n = 50) with metastatic pancreatic ductal adenocarcinoma achieved disease control when treated with napabucasin, nab-paclitaxel, and gemcitabine (Table 1) [78]. This included a 4% complete response rate (n = 2) and a 52% overall disease control rate (n = 26) (NCT02231723) [78]. Unfortunately, no improvement in OS was observed in the napabucasin treatment arm, resulting in trial termination (NCT02993731) [79]. Conversely, the status of a Phase III trial assessing the combination of napabucasin with FOLFIRI versus napabucasin alone for mCRC in patients who experienced treatment failure with chemotherapy regimens remains unpublished (NCT03522649) [76].

TTI-101 is a novel STAT3 inhibitor currently undergoing an active Phase I dose-escalation clinical trial for advanced gastrointestinal solid tumors (NCT03195699) [81]. An ongoing single-arm trial is currently exploring the disease control rate of AZD9150, an antisense oligonucleotide STAT3 inhibitor, in combination with durvalumab. This trial is focused on patients with pancreatic cancer and mismatch-repair-deficient (dMMR) mCRC (NCT02983578) [82].

An alternative to blocking antidifferentiation signals is to enhance pro-differentiation signals. To achieve this, trials are being performed using MTL-C/EBPα, an encapsulated small activating ribonucleic acid (saRNA) encoding pro-differentiation transcription factor C/EBPα (NCT04710641).

### 8.2. All-Trans Retinoic Acid (ATRA)

ATRA, a metabolite of vitamin A, is recognized for its ability to promote cell differentiation and maturation [83]. ATRA is thought to promote the differentiation of MDSCs by increasing intracellular glutathione synthesis and inhibiting reactive oxygen species production [84]. A current clinical trial is investigating HF1K16, an ATRA liposome capable of delivering ATRA to the TME, for tolerability and safety in patients with solid tumors (NCT05388487) [85]. The trial is in progress, currently with an enrollment of 11 patients [85]. A Phase II trial of ATRA and nivolumab in patients with unresectable locally advanced, recurrent, or metastatic pancreatic cancer is recruiting in China (NCT05482451) [86]. The Phase I STAR-PAC trial combining ATRA with gemcitabine-nab-paclitaxel in pancreatic cancer showed safety and improved median overall survival in patients who had received at least two cycles when compared to historical controls in preliminary data [87]. ATRA is also being testing in combination with anti-VEGF mAb (bevacizumab) and anti-PDL1 (atezolizumab) in mCRC (NCT05999812) [88].

### 8.3. Blocking Induction of Immunosuppressive Mediators: Targeting the Interleukin-4 Receptor Alpha (IL4Ra)-STAT6 with Phosphodiesterase (PDE) Inhibitors

Another method of targeting MDSCs is to block the signaling pathways necessary for upregulation of effectors of immunosuppression such as Arginase I and iNOS. Previously, it had been shown that inhibition of PDE5 in purified MDSCs leads to downregulation of IL4Ra, a signaling receptor that drives increase in expression of immunosuppressive Arginase I through STAT6 signaling [89]. PDE5 inhibitors sildenafil and tadalafil have already been FDA-approved for other indications such as erectile dysfunction and pulmonary hypertension, and research has turned towards utilizing these medications in immunotherapy in cancer.

To test whether PDE5 inhibitors (tadalafil) impact treatment in gastric cancers, a trial is underway using tadalafil with or without chemotherapy in resectable gastric cancers (NCT05709574). Tadalafil is being tested in combination with pembrolizumab, ipilimumab, and tumor-associated antigen adjuvant CRS-207 in metastatic pancreatic cancer (NCT05014776) [90].

### 8.4. MDSC Depletion

MDSC depletion from the TME is another strategy to target MDSCs. In tumor-bearing mice, the antimetabolite 5-FU has shown selective induction of MDSC-apoptosis both in vitro and in vivo, enhancing cytotoxic T-cell activation [91]. A current clinical trial is recruiting patients with locally advanced HCC to evaluate the overall response rate of hepatic artery infusion chemotherapy (HAIC) with 5-FU and cisplatin followed by autologous natural killer (NK) cells (Vax-NK/HCC) infusion (NCT050404438) [92]. The researchers hypothesize that 5-FU will deplete MDSCs from the TME, allowing for a more robust antitumor response from the autologous NK cells. Platinum agents have also been thought to target MDSCs.

Another therapeutic target that has been used to facilitate MDSC depletion is gemtuzumab ozogamicin. Gemtuzumab is a CD33-targeting antibody drug conjugate which is FDA-approved in the treatment of acute myeloid leukemia [93]. Human MDSCs express the myeloid marker CD33, making this a good target for therapy. Peripheral blood samples taken from multiple malignancies (lung, prostate, colon, pancreatic, breast) were treated with gemtuzumab, which showed a significant decrease in the expression levels of CD14+CD33+ M-MDSCs with reciprocal proliferation of T-cells, suggesting that repurposing this drug could be a strategy used to target GI malignancies in the future [94]. In addition to CD33, MDSCs have also been shown to upregulate the death receptor 5 (DR5) protein due to endoplasmic reticulum stress. A Phase I clinical trial demonstrated depletion of PMN-MDSCs and early MDSCs in cancer patients following treatment with an agonist antibody to tumor necrosis factor (TNF)-related apoptosis inducing ligand (TRAIL-32) [95]. A Phase I trial used TRAIL-32 antibody as a selective inhibitor of MDSC function, enrolling 16 patients with advanced cancers (colorectal, hepatocellular, or appendiceal carcinoma). The participants were injected with the trial drug every 3 weeks until disease progression or unacceptable toxicity [95]. In addition to favorable safety profile, the trial showed a reversible reduction in MDSCs, with several patients experiencing an MDSC rebound, rationalizing the need for combination immunotherapy [95].

## 9. Inhibition of MDSC Function

### 9.1. Targeting of Arginase I (ARG-1)

The dysregulated metabolism of L-arginine has been shown to play a role in MDSC-related immunosuppression. Under normal physiologic conditions, L-arginine is metabolized in two distinct ways: (1) nitric oxide synthase (iNOS or NOS2) metabolizes L-arginine to L-citrulline and nitric oxide, or (2) arginase metabolizes L-arginine to L-ornithine and urea [96]. Increased expression of ARG-1 by both neoplastic cells and MDSCs is believed to promote the rapid proliferation of cancer cells while simultaneously depleting arginine, which is necessary for proper T-cell function in the TME [97]. Rodriguez et al. demonstrated in a murine lung carcinoma model that ARG-1 decreased the expression of the T-cell receptor CD3ζ, which led to decreased T-cell infiltration, proliferation, and cytokine production [98]. In vitro experiments demonstrated that MDSCs exhibited elevated levels of cationic amino acid transporter 2B (CAT-2B), leading to the depletion of extracellular L-arginine [99]. This depletion hindered the re-expression of CD3ζ and antigen-specific proliferation of activated T-cells, indicating the suppressive effect of MDSCs on T-cell function. A Phase I clinical trial evaluated the arginase-1 peptide vaccine (ARG1-18, 19, 20) in combination with the adjuvant Montanide ISA-51 in advanced refractory solid tumors (including CRC) with a favorable safety profile [100]. In the peripheral blood, peptide-specific immune responses were observed in 90% of patients. INCB001158, the first arginase inhibitor in its class, underwent Phase I/II clinical trials as monotherapy and in combination therapy with pembrolizumab for patients with advanced/metastatic solid tumors [101]. Among the patients with MSS CRC, one patient exhibited a response in the monotherapy group (n = 33), and three demonstrated responses in the combination group (n = 43). Additionally, there was an observed increase in total intratumoral CD8^+^ cells following combination treatment (NCT02903914) [102]. Another study has released preliminary data following an investigation into the combination of INCB001158 with various chemotherapy regimens for advanced and metastatic solid tumors [103]. Javle et al. demonstrated that individuals with biliary tract cancer (BTC) undergoing first-line treatment with gemcitabine plus cisplatin and INCB001158 did not show new safety signals with an ORR of 24% and stable disease in 42% of patients (NCT03314935) [103].

### 9.2. Tryptophan Depletion and Indoleamine 2,3-Dioxygenase (IDO)

Tryptophan is an essential amino acid necessary for proper T-cell function. Enzymes such as IDO-1, IDO-2, and tryptophan 2,3-dioxygenase degrade _L_-tryptophan to N-formyl-L-kynurenine and are overexpressed in cancer cells and MDSCs [104]. IDO expression is also recognized as a mechanism employed by cancer cells to recruit, expand, and activate MDSCs in a Treg-dependent manner [105]. IDO-expressing tumors exhibit heightened aggressiveness and resistance to T-cell-targeting immunotherapies [105]. The selective inhibition of IDO-1 with epacadostat has garnered significant interest in numerous clinical trials due to its potential to reverse immunosuppression. A Phase I/II clinical trial evaluated the safety, tolerability, and maximum tolerated dose (MTD) of epacadostat in Phase I, and the investigator-assessed ORR in Phase II, in combination with pembrolizumab and tumor-appropriate chemotherapy regimens for advanced solid tumors (NCT03361228) [106]. In Phase II, the trial was terminated with no PFS or OS benefit with epacadostat in combination with pembrolizumab compared to the placebo and pembrolizumab arm [106]. In gastroesophageal junction and gastric cancer, a Phase II trial investigating epacadostat with pembrolizumab showed serious adverse events, including jejunal perforation, sepsis, and pleural effusion, with 2/3 patients had disease progression (NCT03196232) [107].

## 10. Reactive Oxygen Species (ROS)

MDSCs may generate reactive oxygen species and peroxynitrite, which modify signaling receptors on the T-cell surface, making them harder to activate [108]. Expression of NOX2, an essential enzyme in driving ROS production, is controlled by MDSC regulator Stat3 [109]. Histamine dihydrochloride has recently been shown to act as an inhibitor of NOX2, and addition of histamine dihydrocholoride boosts responses to immunotherapy in multiple cancer models [110,111]. The Phase I/II PANCEP trial is testing the efficacy of HDC (histamine dihydrochloride) and IL-2 in resectable primary pancreatic adenocarcinoma [112]. The study plans to inject HDC and IL-2 in both peri- and postoperative periods and assess for adverse events as well as changes in myeloid cells and T-cell populations [112]. Other NOX inhibitors are in development but have not yet reached the clinical trial stage.

## 11. Discussion/Future Directions

Malignancies of the gastrointestinal tract are difficult cancers to manage with conventional chemotherapy, surgery, radiation, and immunotherapy. There remains an unmet need to find new targets to improve outcomes. Myeloid-derived suppressor cells are an immunosuppressive cell population that emerging research has shown to play an important role in tumor growth, metastasis, and resistance to treatment. There is a surplus of available clinical data with MDSCs in treatment of metastatic GI malignancies in combination with immunotherapy. Cui et al. have also published a comprehensive review on the pathophysiology of MDSCs in GI cancers [52]. Our review primarily focuses on the clinical aspects of the available therapeutic agents based on the studied MDSCs targets and provides an update to the clinical trial landscape.

Despite the promising preclinical data, in previously treated metastatic colorectal and pancreatic cancer, combination of therapies targeting MDSC migration and differentiation (CCR5, STAT3, CXCR2) have failed to meet endpoints with regards to patient survival. Reasons for this failure remain a topic of investigation but could include additional mechanisms of immunosuppression, recruitment through alterations in tumor microbiome, or a change in flux of MDSCs in late-stage tumors when compared to early-stage tumors. Ongoing trials with agents that can deplete MDSCs from the tumor microenvironment or interfere with MDSC function may have better responses than these initial trials. In addition, future and ongoing clinical work will unravel the efficacy of MDSCs targeting in both the neoadjuvant setting as well as a potential first-line therapy. Genetic and transcriptomic profiling of patients in trials with MDSCs should also be performed to identify biomarkers of good MDSC response. Overall, myeloid-derived suppressor cells represent a promising new avenue of treatment for GI malignancies, one whose therapeutic potential has barely been explored.

## Figures and Tables

**Figure 1 ijms-25-02985-f001:**
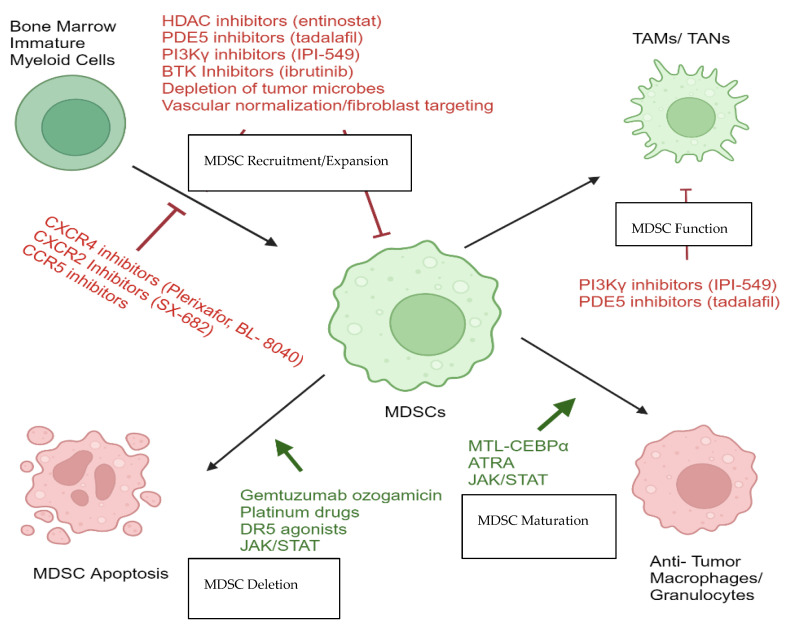
Overview of therapeutic agents specifically designed to target MDSCs in GI malignancies and solid tumors, employing both direct and indirect approaches. Classification of treatment strategies include (1) blocking MDSC recruitment and expansion, (2) promotion of MDSC differentiation, (3) MDSC depletion, and (4) alteration of MDSC metabolic pathways/inhibition of MDSC function, for which more details are given in Section 7 through Section 11.

**Table 1 ijms-25-02985-t001:** Clinical trials targeting MDSCs in gastrointestinal cancer and solid tumors.

Mechanism	Therapeutic Target	Drug	Combination Therapy	Cancer Type	Phase	Status	Trial ID	Response (PFS, OS) in Months)	Safety (% Serious Adverse Events
Blocking MDSC Recruitment and Expansion	CCR5	Maraviroc	-	mCRC (Liver Metastases)	I	Completed	NCT01736813	-	10.6
		Maraviroc	Pembrolizumab	MSS mCRC	I	Completed	NCT03274804	2.1, 9.8	20
		Maraviroc	Ipilimumab and Nivolumab	mCRC and PC	I	Completed	NCT04721301	N/A	N/A
		Maraviroc	Pembrolizumab	mGC	I	Not yet recruiting	ChiCTR-OIC-17013778	-	-
		OB-002		mCRC, PC, GC	I	Not yet recruiting	NCT05940844	-	-
		Vicriviroc	Pembrolizumab	MSS mCRC	II	Completed	NCT03631407	2.1, 4.6	50
		BMS-813160	GVax Cancer Vaccine, Nivolumab	PC	I/II	Active	NCT03767582	-	-
		BMS-813160	Chemotherapy or Nivolumab	mCRC, PC	Ib/II	Completed	NCT03184870	-	-
		BMS-813160	Nivolumab, gemcitabine, and nab-paclitaxel	Borderline resectable or locally advanced PC	I/II	Completed	NCT03496662	-	-
	CXCR1/2	SX-682	Nivolumab	RAS-mutated, MSS CRC	Ib/II	Recruiting	NCT04599140	-	-
		SX-682	Tislelizumab	PC	II	Recruiting	NCT05604560	-	-
		SX-682	Nivolumab	mPC	I	Recruiting	NCT04477343	-	-
		SX-682	BA and CV301	PBC	I/II	Active, not recruiting	NCT04574583	-	-
		AZD5069	Durvalumab	HCC	I/II	Active	ISRCTN12669009	-	-
		AZD5069	Durvalumab	mPC	Ib/II	Completed	EUCTR2015-003639-37-GB	1.6, NA	2.8
	PI3K	Duvelisib	SG001 (anti-PD1)	Solid tumors	I/II	Not Yet Recruiting	NCT05508659	-	-
		ZX-4081	None	Advanced Solid Tumors	I	Recruiting	NCT05118841	-	-
		ZX-101A	-	Advanced solid tumors	I/II	Not yet recruiting	NCT05258266	-	-
	PDE	Tadalafil	Tadalafil and FLOT	Stage I–III GC/GEJ	II	Recruiting	NCT05709574	-	-
Promotion of MDSC differentiation	ATRA analogues	HF1K16	-	Solid tumors (including colon cancer)	I	Recruiting	NCT05388487	-	-
	STAT3	Napabucasin	Nivolumab	MSS mCRC	II	Completed	NCT03647839	-	-
		TTI-101	-	HCC, CRC GC	I	Active, not recruiting	NCT03195699	-	-
		Napabucasin	FOLFIRI	mCRC	III	Unknown status	NCT03522649	-	-
		AZD9150	Durvalumab	mPC, dMMR mCRC	II	Active, not recruiting	NCT02983578	-	-
		AZD9150	-		II	Terminated	NCT02417753	-	-
		SC-43	Cisplatin	BTC	I/II	Unknown status	NCT04733521	-	-
		Napabucasin	Nab-paclitaxel and gemcitabine	mPDAC	III	Completed	NCT02993731 NCT02231723	6.7, 11.4	59
MDSC Depletion	MDSCs	Vax-NK/HCC	5-FU	Locally advanced HCC	IIa	Recruiting	NCT050404438	-	-
Alteration of MDSC metabolic pathways	Arginase-1	ARG1-18, 19, 20	Adjuvant Montanide ISA-51	CRC (Metastatic Solid Tumors)	I	Completed	NCT03689192	2.0, 7.3	30
		INCB001158	Pembrolizumab	CRC, GC (Advanced/Metastatic Solid Tumors)	I/II	Completed	NCT02903914	-	36.5
		INCB001158	Epacadostat Pembrolizumab	Advanced/Metastatic Solid Tumors	I/II	Terminated	NCT03361228	-	-
		INCB001158	FOLFOX/Gemcitabine + Cisplatin/Paclitaxel	CRC, BTC, GEC (Advanced/Metastatic Solid Tumors)	I/II	Completed	NCT03314935	variable	55
		INCB001158	Epacadostat Pembrolizumab	Advanced/Metastatic Solid Tumors	I/II	Terminated	NCT03361228	-	-
	IDO-1	Epacadostat	Pembrolizumab and Chemotherapy	CRC, AC, CC, EC, GC, gastric, GEJ, PC	I/II	Completed	NCT03085914	4.7, 31.4	78.6
		Epacadostat	Short-course radiation + CAPOX chemotherapy + surgery	RC	I/II	Recruiting	NCT03516708	-	-
		Epacadostat	Pembrolizumab	GEJ, GC	II	Completed	NCT03196232	0, 33	-
		Epacadostat	Pembrolizumab	GEJ/EC	II	Terminated	NCT03592407	-	-
		BMS-986205	Nivolumab	Unresectable mHCC	I/II	Terminated	NCT03695250	-	-
Inhibition of MDSC function	Trail-R2	DS-8273a	-	CRC, PC	I	completed	NCT02076451		
	NOX1	HDC/IL2	-	PC	I/II	Recruiting	NCT05810792	-	-

CCR5 = cysteine-cysteine motif chemokine receptor 5, CRC = colorectal cancer, mCRC = metastatic colorectal cancer, MSS = microsatellite stable, dMMR = deficient mismatch repair, PC = pancreatic cancer, ATRA = all-trans retinoic acid, HCC = hepatocellular carcinoma, 5-FU = 5-fluorouracil, GEJ = gastroesophageal junction, GC = gallbladder cancer, mPDAC = metastatic pancreatic ductal carcinoma, BTC = biliary tract carcinoma, CC = cholangiocarcinoma, AC = anal cancer, EC = esophageal cancer, RC = rectal cancer, GC = gastric cancer, mPC = metastatic pancreatic cancer, PBC = pancreaticobiliary cancer, mHCC = metastatic hepatocellular cancer, CXCR1/2 = C-X-C receptor 1/2, PI3K = Phosphoinositide 3-kinase, VEGF = vascular endothelial growth factor, IL-2 = interleukin 2, IL-1B = interleukin 1 beta, PDE = phosphodiesterase, STAT3 = Signal transducer and activator of transcription 3, IDO-1 = Indoleamine 2, 3-dioxygenase-1, Trail-R2 = tumor necrosis factor–related apoptosis–inducing ligand receptor 2, NOX1 = NADPH oxidase 1, ARG1 = arginase 1, HDC = histone deacetylase, CAPOX = capecitabine, oxaliplatin, FLOT = 5Florouracil, oxaliplatin, docetaxel, FOLFIRI = 5Florouracil, Irinotecan.

## Data Availability

Not applicable.

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
