# Peer review of "Myeloid-Derived Suppressor Cells: Therapeutic Target for Gastrointestinal Cancers"

_ijms, 2024, doi:10.3390/ijms25052985_

Round 1

Reviewer 1 Report

Comments and Suggestions for Authors

The manuscript titled "Myeloid-Derived Suppressor Cells: Therapeutic Target for Gastrointestinal Cancers" by Junaid Arshad et al. provides a comprehensive and insightful exploration of the challenges associated with treating gastrointestinal cancers. The authors effectively highlight the limited success of current strategies and emphasize the importance of understanding the tumor microenvironment for the development of novel therapeutics. The focus on myeloid-derived suppressor cells (MDSCs) as crucial drivers in the pathogenesis and progression of gastrointestinal malignancies is particularly commendable. However, it is noteworthy that the manuscript currently requires further adjustments and improvements before it can be considered for publication.

The following suggestions and corrections aim to enhance the overall quality and readability of the manuscript.

General Comments: 

I noticed that the references are not consistently formatted. Please ensure that the references are numbered consistently and that the reference numbers appear within square brackets, such as [1], [2], etc. This will enhance the overall clarity and uniformity of the citation style throughout the manuscript.

Review the document for any grammatical errors or typos to ensure a polished and error-free presentation.

Consider breaking down the lengthy paragraphs into smaller, focused sections. This will enhance readability and facilitate better comprehension for readers.

Some statements lack in-text citations. Ensure that every factual claim or statement is supported by appropriate references.

It might be beneficial to include a brief sentence in Introduction that outlines the importance of understanding the TME and MDSCs in the context of developing more effective anti-cancer therapies.

Some sentences and phrases are repeated throughout the document. Ensure that each piece of information is presented concisely, avoiding unnecessary redundancy.

Consider adding a brief concluding section that summarizes the key findings and their implications. This will provide closure to the reader and reinforce the significance of the discussed topics.

Table 1 requires references; please modify the use of "+" to "and".

It would be appreciated if you could provide Figure 1 with improved clarity. Additionally, the title of Figure 1 does not adequately describe its content.

The manuscript requires a section where author contributions are clearly described.

The references section needs to be overhauled to achieve improved organization. If a reference includes a suitable DOI, there is no necessity to add a separate link. Additionally, discrepancies exist where some DOIs are associated with inappropriate references. Moreover, there is an inconsistency in the total number of references between the text (116) and the reference section (106). Furthermore, the sequence of references in the text is not consistently reflected in the reference section.

Comments on the Quality of English Language

Overall, the manuscript needs moderate english editing. The authors are asked to review the document for any grammatical errors or typos to ensure a polished and error-free manuscript.

Author Response

I noticed that the references are not consistently formatted. Please ensure that the references are numbered consistently and that the reference numbers appear within square brackets, such as [1], [2], etc. This will enhance the overall clarity and uniformity of the citation style throughout the manuscript.

  • The citation style is changed to that of the journal.

Review the document for any grammatical errors or typos to ensure a polished and error-free presentation.

  • The relevant changes have been made.

Consider breaking down the lengthy paragraphs into smaller, focused sections. This will enhance readability and facilitate better comprehension for readers.

  • Based on this feedback, we have done extensive revisions to improve clarity.

Some statements lack in-text citations. Ensure that every factual claim or statement is supported by appropriate references.

  • The relevant changes have been made to avoid uncited statements.

It might be beneficial to include a brief sentence in Introduction that outlines the importance of understanding the TME and MDSCs in the context of developing more effective anti-cancer therapies.

  • We have added a sentence in the introduction that specifically addresses this issue.

Some sentences and phrases are repeated throughout the document. Ensure that each piece of information is presented concisely, avoiding unnecessary redundancy.

  • We have made revisions in the article to make it more concise and eliminate unnecessary sentences.

Consider adding a brief concluding section that summarizes the key findings and their implications. This will provide closure to the reader and reinforce the significance of the discussed topics.

  • We have tried to stay broad in the conclusion, trials are all in early stages (I/II). It is hard to draw significant conclusions regarding key findings and implications. We left it open for readers to draw conclusion.

Table 1 requires references; please modify the use of “+” to “and”. (ADDRESSED)

  • Referenced the trials in the paper text, the table is provided as a more concise reference.

It would be appreciated if you could provide Figure 1 with improved clarity. Additionally, the title of Figure 1 does not adequately describe its content.

  • We have modified the figure legend and made changes in the figure to improve clarity.

The manuscript requires a section where author contributions are clearly described.

  • Authors contribution section is added.

The references section needs to be overhauled to achieve improved organization. If a reference includes a suitable DOI, there is no necessity to add a separate link. Additionally, discrepancies exist where some DOIs are associated with inappropriate references. Moreover, there is an inconsistency in the total number of references between the text (116) and the reference section (106). Furthermore, the sequence of references in the text is not consistently reflected in the reference section.

  • We have fixed the references in consistent with the journal style and preferences. We have also fixed the discrepancies.

Reviewer 2 Report

Comments and Suggestions for Authors

While the article provides valuable insights into the role of MDSCs in GI cancers, several aspects are not addressed. Please include the answers to these questions in the article text if applicable.

1. Although the role of MDSCs in GI cancers is covered in great detail in the article, it does not explicitly address the implications for treatment strategies of a detailed analysis of their differential expression across different types of GI cancers, such as colorectal, gastric, and pancreatic cancers. Given the heterogeneity of these cancers, it is imperative to comprehend this aspect as it may contribute to the development of more tailored and efficient treatment strategies. Has research been done on how different types of gastrointestinal cancers may differ in the expression or activity of MDSCs, and how this might impact treatment approaches?

2. I found the section on immune checkpoints very insightful. However, I'm interested to know more about how MDSCs specifically interact with these checkpoints. Could this knowledge potentially refine our strategies in using checkpoint inhibitors alongside MDSC-targeted treatments in GI cancers?

3. The function of MDSCs in the tumor microenvironment is excellently described in the article, but I'm not clear on the precise molecular pathways that these cells follow. Could you provide some information about these pathways and how they might be targeted to stop MDSCs from promoting tumors?

4. The idea of specifically targeting MDSCs while avoiding harm to other myeloid cells intrigues me. The difficulties in obtaining this selectivity—such as figuring out the distinct molecular markers of MDSCs or creating agents with different effects on MDSCs—are not discussed in the article. What obstacles need to be overcome to create pharmaceuticals that selectively target MDSCs while avoiding affecting the larger myeloid cell population?

5. There is no information in the article about clinical trials that examine how the gut microbiota affects MDSC activity. Such knowledge could be essential to comprehending and possibly regulating MDSC activity in GI cancers, given the mounting evidence of the microbiome's impact on cancer progression and treatment response. Do there exist any clinical trials that are looking into how the gut microbiota affects MDSC activity and how that affects the progression of GI cancer?

6. The article skips over developments in biomedical engineering that might help identify and describe MDSCs. These technologies are essential for tracking treatment responses and increasing the accuracy of diagnoses, particularly in the context of personalized medicine. What developments in biomedical engineering are being investigated to more accurately identify and describe MDSCs in the context of GI cancer, such as imaging or biomarker detection?

7. While the article reviews the role of MDSCs in GI cancers, it does not detail their morphological and phenotypic characteristics across different cancer stages. For diagnostic and prognostic purposes, pathologists require this information, which may help with more precise staging and treatment planning. What are the implications for diagnosis and prognosis when it comes to the morphological and phenotypic characteristics of MDSCs varying between different stages of GI cancers?

8. The aspects of developing MDSC-targeted therapies related to the market are not covered in the article. Pharmaceutical professionals must comprehend these obstacles and opportunities to successfully negotiate the commercial landscape of cancer therapeutics. What are the possible obstacles and prospects in the market for creating MDSC-targeted treatments for gastrointestinal cancers?

9. The article skips over the implications of introducing MDSC-targeted therapies for healthcare policy and the economy. Understanding the cost-effectiveness and resource allocation for these novel therapies is critical for administrators and policymakers, particularly as healthcare systems around the world shift toward more individualized medical practices. What effects do MDSC-targeted treatments have on the utilization of healthcare resources and costs, especially when it comes to GI cancer personalized medicine?

10. While an outline of MDSCs' role in gastrointestinal cancers is given in the article, neither the underlying mechanisms nor their particular contribution to treatment resistance are thoroughly examined. Closing this gap may help discover new ways to combat resistance, like combination therapies that target MDSCs in addition to traditional treatments. Comprehending these mechanisms is also essential for creating prognostic biomarkers for resistance to treatment and customizing therapies for each patient. What possible roles might MDSCs play in the therapeutic resistance context when it comes to the ineffectiveness of current treatments for gastrointestinal cancers, and what possible mechanisms might underlie this phenomenon? Additionally, does the article explore any strategies to overcome this resistance mediated by MDSCs?

Author Response

  1. Although the role of MDSCs in GI cancers is covered in great detail in the article, it does not explicitly address the implications for treatment strategies of a detailed analysis of their differential expression across different types of GI cancers, such as colorectal, gastric, and pancreatic cancers. Given the heterogeneity of these cancers, it is imperative to comprehend this aspect as it may contribute to the development of more tailored and efficient treatment strategies. Has research been done on how different types of gastrointestinal cancers may differ in the expression or activity of MDSCs, and how this might impact treatment approaches?

-           We thank the reviewer for this feedback. As far as we know, there hasn’t been a head-to-head pan-cancer comparison of MDSCs in GI malignancies. Unlike with T cells, there has not been clear data showing heterogeneity in MDSC activity across cancer, outside of the aforementioned G-MDSC and M-MDSC classifications. While there is some variation in myeloid infiltration across GI malignancies (Cheng S et al 2021, Cell), it is still unclear whether this is biologically significant with regards to treatment approaches. Due to the above reasons, this was not included in the current review

  1. I found the section on immune checkpoints very insightful. However, I’m interested to know more about how MDSCs specifically interact with these checkpoints. Could this knowledge potentially refine our strategies in using checkpoint inhibitors alongside MDSC-targeted treatments in GI cancers?

-           Thank you for this thoughtful question. We have expanded the discussion lines 57-65 (original manuscript) with more information about how MDSCs do immunosuppression. Briefly, MDSCs can act to suppress T cell responses through expression of checkpoint molecules on MDSCs themselves (i.e. PD-L1), or act to suppress T cell proliferation and activity downstream of this. From clinical practice, we know that GI malignancies do not respond to immunotherapy, suggesting mechanisms of resistance. As MDSCs can suppress T cell responses from multiple mechanisms outside of the classical immune checkpoints, this suggests that in tumors with a high intra-tumoral MDSC infiltration, we may need to combine classical immune checkpoint blockade with MDSC targeted therapies in order to get more effective immune responses.

  1. The function of MDSCs in the tumor microenvironment is excellently described in the article, but I’m not clear on the precise molecular pathways that these cells follow. Could you provide some information about these pathways and how they might be targeted to stop MDSCs from promoting tumors?

-           Thank you for raising this point. The basic immunosuppressive mechanisms of MDSCs have been added in Section 2, lines 57-65 (original manuscript). The discussion regarding the pathways involved in recruitment and differentiation is in sections 7-11 of the revised manuscript.

  1. The idea of specifically targeting MDSCs while avoiding harm to other myeloid cells intrigues me. The difficulties in obtaining this selectivity—such as figuring out the distinct molecular markers of MDSCs or creating agents with different effects on MDSCs—are not discussed in the article. What obstacles need to be overcome to create pharmaceuticals that selectively target MDSCs while avoiding affecting the larger myeloid cell population?

-           Thank you for this point. The larger myeloid cell population is quite heterogeneous, including neutrophils, dendritic cells, and macrophages. Most MDSCs don’t have a cell surface marker that differentiates them from more differentiated populations like neutrophils and macrophages; hence, often therapies targeting PMN-MDSCs will hit neutrophils and therapies targeting M-MDSC often affect monocytes and macrophages. We have discussed cell surface markers in the section 2. This probably is one of the biggest obstacles to therapies you envision. Even in humans, they are defined by multiple markers. Perhaps a bispecific antibody or engineered cell receptor could work. However, hitting differentiated populations may not necessarily be a bad thing as tumor-associated macrophages or neutrophils also have immunosuppressive roles in various cancers.

  1. There is no information in the article about clinical trials that examine how the gut microbiota affects MDSC activity. Such knowledge could be essential to comprehending and possibly regulating MDSC activity in GI cancers, given the mounting evidence of the microbiome’s impact on cancer progression and treatment response. Do there exist any clinical trials that are looking into how the gut microbiota affects MDSC activity and how that affects the progression of GI cancer?

-           Thank you for raising this point. Unfortunately, research on the role of the microbiota in GI malignancies is still in its infancy, so there isn’t a lot of info on this. There are no clinical trials yet looking into how gut microbiota affect progression of GI cancer. There have been isolated case reports showing early use of antibiotics in life can increase risk of GI malignancies (Yan C et al 2019, World J Clinical Cases), perhaps through microbial dysbiosis, but no one has tested if altering microbiota affect progression of GI cancers. We have included one relevant clinical trial using the vancomycin with tadalafil and nivolumab in hepatocellular carcinoma (Reference 78). The trial failed to show encouraging results.

  1. The article skips over developments in biomedical engineering that might help identify and describe MDSCs. These technologies are essential for tracking treatment responses and increasing the accuracy of diagnoses, particularly in the context of personalized medicine. What developments in biomedical engineering are being investigated to more accurately identify and describe MDSCs in the context of GI cancer, such as imaging or biomarker detection?

-           We thank the reviewer for their concerns. While the topic of new technologies to identify MDSCs is interesting, we feel it is outside the scope of this paper which is focused on the function and targeting of MDSCs in GI malignancies.

  1. While the article reviews the role of MDSCs in GI cancers, it does not detail their morphological and phenotypic characteristics across different cancer stages. For diagnostic and prognostic purposes, pathologists require this information, which may help with more precise staging and treatment planning. What are the implications for diagnosis and prognosis when it comes to the morphological and phenotypic characteristics of MDSCs varying between different stages of GI cancers?

-           We have discussed some of the functions of MDSCs in the revised document section 2. Furthermore, in lines 48-59 (original manuscript), we discuss some of the pathologic designations of MDSCs (i.e. PMN-MDSC and M-MDSC). The distinct functions of these different subsets of MDSCs in various GI malignancies is discussed throughout the paper. We hope this provides some clarification to the authors’ concerns.

  1. The aspects of developing MDSC-targeted therapies related to the market are not covered in the article. Pharmaceutical professionals must comprehend these obstacles and opportunities to successfully negotiate the commercial landscape of cancer therapeutics. What are the possible obstacles and prospects in the market for creating MDSC-targeted treatments for gastrointestinal cancers?

-           We appreciate the author’s concerns, but as this paper’s focus is on the function of MDSCs in GI malignancies and targeting. We feel that this topic, while fascinating, is outside the scope of this paper and may be a focus for future work.

  1. The article skips over the implications of introducing MDSC-targeted therapies for healthcare policy and the economy. Understanding the cost-effectiveness and resource allocation for these novel therapies is critical for administrators and policymakers, particularly as healthcare systems around the world shift toward more individualized medical practices. What effects do MDSC-targeted treatments have on the utilization of healthcare resources and costs, especially when it comes to GI cancer personalized medicine?

-           We acknowledge the author’s concerns, but as this paper’s focus is on the function of MDSCs in GI malignancies with therapeutic targeting. We feel that healthcare utilization, cost effectiveness and resource allocation for MDSC targeted therapies is outside the scope of this paper and can be considered for future projects.

  1. While an outline of MDSCs’ role in gastrointestinal cancers is given in the article, neither the underlying mechanisms nor their particular contribution to treatment resistance are thoroughly examined. Closing this gap may help discover new ways to combat resistance, like combination therapies that target MDSCs in addition to traditional treatments. Comprehending these mechanisms is also essential for creating prognostic biomarkers for resistance to treatment and customizing therapies for each patient. What possible roles might MDSCs play in the therapeutic resistance context when it comes to the ineffectiveness of current treatments for gastrointestinal cancers, and what possible mechanisms might underlie this phenomenon? Additionally, does the article explore any strategies to overcome this resistance mediated by MDSCs?

-           Thank you for these comments. We have revised original document section 6 (lines 111-161) with more of a focus on the mechanisms that drive recurrence after conventional therapy. Our work mentions that certain chemotherapy regimens such as FOLFOX tend to have less MDSC-inductive effects when compared to others like FOLFIRI, so this is one strategy that could be tried to reduce MDSC infiltration in tumors. Notably, FOLFOX contains oxaliplatin, which is thought to be a mediator of immunogenic cell death, which may explain the differences in responsiveness. Edited version section 7 and onwards discuss strategies for targeting MDSCs, including decreasing MDSC recruitment to the tumor microenvironment, depletion, and reprogramming.

Reviewer 3 Report

Comments and Suggestions for Authors

The review is comprehensive, but heavy to read as in many places it is simply just lists of data from publications without any further insight. Therefore, the authors should demonstrate their expertise by concluding the existing literature at the end of different chapters/paragraphs. Also, an additional figure or two would make the reading more enjoyable.

Table: It would be informative, if the outcomes (response, safety) of the trials are included in those cases this information is available.

The abbreviations should be spelled out or explained. For example, what is MTL-CEBPalpha. Also, the names of antibody/drug targets should be given. Only few readers may know what is Tislelizumab etc…

Minor:

The terminology should be uniform, for example CD instead of Cd and not variable.

Page 7, line 201 CCR-CCL5; should be CCR5???

Author Response

The review is comprehensive, but heavy to read as in many places it is simply just lists of data from publications without any further insight. Therefore, the authors should demonstrate their expertise by concluding the existing literature at the end of different chapters/paragraphs. Also, an additional figure or two would make the reading more enjoyable.

  • Thank you for this comment. We have repurposed sections of the physiology in the first half and the second half of the paper.

Table: It would be informative, if the outcomes (response, safety) of the trials are included in those cases this information is available.

  • Thank you for this comment. We edited the table to reflect this in trials where data was available.

The abbreviations should be spelled out or explained. For example, what is MTL-CEBPalpha. Also, the names of antibody/drug targets should be given. Only few readers may know what is Tislelizumab etc…

  • These issues are addressed

Minor:

The terminology should be uniform, for example CD instead of Cd and not variable.

  • Thank you for this comment. We have reviewed the manuscript again and made these changes to ensure consistency.

Page 7, line 201 CCR-CCL5; should be CCR5???

-           Thank you for the input, we have corrected this error

Reviewer 4 Report

Comments and Suggestions for Authors

This review paper describes the role of myeloid derived suppressor cells as therapeutic target for gastrointestinal cancer. The review gives a thorough overview of the available physiologic background, the targets, the in vitro studies, and the clinical trials addressing this topic. It is very well written.

Still, there are some issues that need to be addressed.

First, this is not the first review on this topic, and especially the sections 2 through 7 have been covered in details elsewhere. Even sections 8 through 11 have. For example, Cui C, Lan P, Fu L. The role of myeloid-derived suppressor cells in gastrointestinal cancer. Cancer Communications. 2021;41(6):442- 504, doi: 10.1002/cac2.12156, did cover exactly the same topic extensively. Now that is a review from 3 years back, and meanwhile more clinical trials have been on-going and started. It is fair to dedicate a paragraph to how that review and the current differ, or what new information the current paper gives. There needs to be a justification given in the current review. Added here could be an explanation on the structure of the current review, for example that the possible therapeutic treatment/drug and its physiologic basis are explained in Figure 1, with more details given in the respective sections of 8 through 10, while clinical trials supposed to provide evidence are listed in Table 1 and described in the same sections. It is suggested to use Figure 1 in the first part of the manuscript (sections 2-7), while Table 1 in the second part of the manuscript (sections 8-11) and adjust the text for the figure.

Second, many sections start with physiology statements that are unreferenced. And during the sections many statements are not references, or give a reference to either the Table or the Figure (were needed). Thus, the citing and the integration of the Table and Figure in the main body of the text should be better performed.

Third, many abbreviations are undefined (section 2-7), or only later on in the paper (section 8-11). While many abbreviations are obvious for someone of the field, it would be good to please a general audience and add small to the point definitions upon first mention of the abbreviation.

Abstract

Lines 12/13: omit ‘in the world ’

Lines 13/14 should read: ‘Current strategies to cure and control gastrointestinal (GI) cancers, like surgery, radiation, chemotherapy, and immunotherapy, have met with limited success,’

Lines 15/20 should read: ‘Myeloid derived suppressor cells (MDSCs) have emerged as crucial drivers of pathogenesis and progression within the tumor microenvironment in GI malignancies. Many MDSCs clinical targets have been defined in preclinical models, that potentially play an integral role in blocking MDSC recruitment and expansion, promoting MDSC differentiation into mature myeloid cells, depleting existing MDSCs, altering MDSC metabolic pathways, and directly inhibiting MDSC function.’

Adjust the keywords.

Give an abbreviation list.

Introduction

Line 52 should read: ‘MDSCs are defined by cells expressing cell surface markers Cd11b+…’

Line 55 should read: ‘In humans, M-MDSCs are defined as expression of cell markers CD14’

Lines 61/64, give references for each of the statements.

Line 66/69 needs a reference of the mouse model.

Lines 69-94 define CXCR2, chemokine receptor and role.

Line 69 define TLR4, toll-like receptor and role.

Lines 83/86 need references for each of the statements.

Line 93 define CCL2, chemoattractant protein and role.

Lines 112/113 give references for ‘MDSCs have been shown to play an important role in regrowth of colorectal, hepatocellular (HCC), and pancreatic cancer.’

Line 121 should read ‘secretion of the cytokine alpha interferon’

Line 135 should read ‘CRC patients treated with a combination of chemotherapy drugs FOLFOX (5FU, Oxaliplatin)’

Line 135 and 140 define 5FU, 5-fluorouracil.

Line 155 define ‘PD-1’ and should read ‘efficacy of anti PD-1 antibody therapy’

Line 163 define ‘ApoE+’, Apolipoprotein E gene and ‘PD-L1’, programmed death-ligand 1 a transmembrane protein or antibody.

Many recent references (>2020) till section 7.

Table 1. Note 1: subscript should include the defined abbreviations: CXCR1/2; PI3K; VEGF; IL-1Beta; PDE; SIRPalpha; STAT3; IDO-1; Trail-R2; NOX1. Also try to have the trial # printed over 1 row only (widening the column). Note 2: include a reference to the 5 treatment strategies of Figure 1.

Line 188/192 should read ‘Figure 1. Overview of therapeutic agents specifically designed to target MDSCs in GI malignancies and solid tumors, employing both direct and indirect approaches. Classification of treatment strategies include: 1) blocking MDSC recruitment and expansion, 2) promotion of MDSC differentiation, 3) MDSC depletion, 4) alteration of MDSC metabolic pathways, and 5) inhibition of MDSC function, for which more details are given in sections 8 through 11. Note 1: Excluding the reference, or mentioning after 6. Note 2: The 5 treatment strategies can be indicated with the respective numbers next to the green- and red- printed agents Note 3: abbreviations should be defined.

Line 195/197 could use references especially for RANTES, and it also the first sentence in which G-protein coupled receptor and chemokine ligands are mentioned. These definition should have been first mentioned earlier.

Line 207: ‘In the MACRON-001 Phase I trial’

Lines 212/215 need the references, and should be identical to the trial# of Table 1.

Line 229 when indicated ‘The CXCLs-CXCR1/2 axis’ in Figure 1, the identification number and figure could be cited in this sentence. Please integrate refer in the main body of the text to your table and especially the Figure.

Line 229/230 needs references.

Lines 236/239 need references ‘Similarly, ... neoadjuvant treatment.’

Lines 239/242 need references ‘Simultaneously, ... first-line regimen.’

Line 267/268 needs references ‘A dose-finding … stromal tumor (GIST).’ Probably (74), but it should be mentioned in this first line.

Line 292/293 needs references in according with the table ‘Other novel … solid tumors (NCT05258266).‘

Line 353 give the reference (87) to the MODULATE study earlier, e.g., line 346.

Lines 353/361 need references if possible ‘TTI-101 … mCRC (NCT02983578).

Lines 372/376 need references ‘A current clinical … NL cell.’

Section 10.1 is well referenced, but the first paragraph is fairly old literature (<2009).

Lines 442/445 need references ‘A current trial … and surgery (NCT03516708)’.

Lines 459/463 need references ‘Another preliminary study … oxidase (NOX2)’.

Author Response

First, this is not the first review on this topic, and especially the sections 2 through 7 have been covered in details elsewhere. Even sections 8 through 11 have. For example, Cui C, Lan P, Fu L. The role of myeloid-derived suppressor cells in gastrointestinal cancer. Cancer Communications. 2021;41(6):442- 504, doi: 10.1002/cac2.12156, did cover exactly the same topic extensively. Now that is a review from 3 years back, and meanwhile more clinical trials have been on-going and started.  It is fair to dedicate a paragraph to how that review and the current differ, or what new information the current paper gives. There needs to be a justification given in the current review.

  • We acknowledge this comment and justification of the article is added in the discussion section.

Added here could be an explanation on the structure of the current review, for example that the possible therapeutic treatment/drug and its physiologic basis are explained in Figure 1, with more details given in the respective sections of 8 through 10, while clinical trials supposed to provide evidence are listed in Table 1 and described in the same sections. It is suggested to use Figure 1 in the first part of the manuscript (sections 2-7), while Table 1 in the second part of the manuscript (sections 8-11) and adjust the text for the figure.

  • We have edited the article to blend the physiology and the clinical trials into broader sections, the Table and Figure should be used as a quick reference and have been edited.

Second, many sections start with physiology statements that are unreferenced. And during the sections many statements are not references or give a reference to either the Table or the Figure (were needed). Thus, the citing and the integration of the Table and Figure in the main body of the text should be better performed.

  • Citations were edited and focus was made on referencing physiology statements that were not common knowledge statements.

Third, many abbreviations are undefined (section 2-7), or only later on in the paper (section 8-11). While many abbreviations are obvious for someone of the field, it would be good to please a general audience and add small to the point definitions upon first mention of the abbreviation.

  • We have made our best efforts to define all the abbreviations.

Abstract

Lines 12/13: omit ‘in the world ’

  • We thank the reviewers for this comment. This was fixed

Lines 13/14 should read: ‘Current strategies to cure and control gastrointestinal (GI) cancers, like surgery, radiation, chemotherapy, and immunotherapy, have met with limited success,’

  • We thank the reviewers for this comment. This was fixed

Lines 15/20 should read: ‘Myeloid derived suppressor cells (MDSCs) have emerged as crucial drivers of pathogenesis and progression within the tumor microenvironment in GI malignancies. Many MDSCs clinical targets have been defined in preclinical models, that potentially play an integral role in blocking MDSC recruitment and expansion, promoting MDSC differentiation into mature myeloid cells, depleting existing MDSCs, altering MDSC metabolic pathways, and directly inhibiting MDSC function.’

  • We thank the reviewers for this comment. This was fixed.

Adjust the keywords.

  • The key words were adjusted.

Give an abbreviation list.

  • All the abbreviations were defined as needed in the article.

Introduction

Line 52 should read: ‘MDSCs are defined by cells expressing cell surface markers Cd11b+…’ 

  • This was addressed in the paper.

Line 55 should read: ‘In humans, M-MDSCs are defined as expression of cell markers CD14’

  • This was addressed in the paper.

Lines 61/64, give references for each of the statements.

  • This was addressed in the paper.

Line 66/69 needs a reference of the mouse model.

  • This was addressed in the edits.

Lines 69-94 define CXCR2, chemokine receptor and role.

  • This was addressed in the edits.

Line 69 define TLR4, toll-like receptor and role.

  • This was addressed in the edits.

Lines 83/86 need references for each of the statements.

  • This was addressed in the edits.

Line 93 define CCL2, chemoattractant protein and role.

  • This was addressed in the edits.

Lines 112/113 give references for ‘MDSCs have been shown to play an important role in regrowth of colorectal, hepatocellular (HCC), and pancreatic cancer.’

  • This was addressed in the edits.

Line 121 should read ‘secretion of the cytokine alpha interferon

  • This was addressed in the edits.

Line 135 should read ‘CRC patients treated with a combination of chemotherapy drugs FOLFOX (5FU, Oxaliplatin)’

  • This was addressed in the edits.

Line 135 and 140 define 5FU, 5-fluorouracil.

  • This was addressed in the edits.

Line 155 define ‘PD-1’ and should read ‘efficacy of anti PD-1 antibody therapy’

  • This was addressed in the edits.

Line 163 define ‘ApoE+’, Apolipoprotein E gene and ‘PD-L1’, programmed death-ligand 1 a transmembrane protein or antibody.

  • This was addressed in the edits.

Table 1. Note 1: subscript should include the defined abbreviations: CXCR1/2; PI3K; VEGF; IL-1Beta; PDE; SIRPalpha; STAT3; IDO-1; Trail-R2; NOX1. Also try to have the trial # printed over 1 row only (widening the column). Note 2: include a reference to the 5 treatment strategies of Figure 1.

  • This was addressed in the edits.

Line 188/192 should read ‘Figure 1. Overview of therapeutic agents specifically designed to target MDSCs in GI malignancies and solid tumors, employing both direct and indirect approaches. Classification of treatment strategies include: 1) blocking MDSC recruitment and expansion, 2) promotion of MDSC differentiation, 3) MDSC depletion, 4) alteration of MDSC metabolic pathways, and 5) inhibition of MDSC function, for which more details are given in sections 8 through 11.’

Note 1: Excluding the reference or mentioning after 6.

Note 2: The 5 treatment strategies can be indicated with the respective numbers next to the green- and red- printed agents

Note 3: abbreviations should be defined.

  • These changes have been made.

Line 195/197 could use references especially for RANTES, and it also the first sentence in which G-protein coupled receptor and chemokine ligands are mentioned. These definition should have been first mentioned earlier.

  • The references have been included.

Line 207: ‘In the MACRON-001 Phase I trial’

  • This change has been included.

Lines 212/215 need the references and should be identical to the trial# of Table 1.

  • The references have been included.

Line 229 when indicated ‘The CXCLs-CXCR1/2 axis’ in Figure 1, the identification number and figure could be cited in this sentence. Please integrate refer in the main body of the text to your table and especially the Figure.

  • This change has been made

Line 229/230 needs references.

  • The references have been added.

Lines 236/239 need references ‘Similarly, ... neoadjuvant treatment.’

  • The reference is added.

Lines 239/242 need references ‘Simultaneously, ... first-line regimen.’

  • The reference is added.

Line 267/268 needs references ‘A dose-finding … stromal tumor (GIST).’ Probably (74), but it should be mentioned in this first line.

  • This section was removed.

Line 292/293 needs references in according with the table ‘Other novel … solid tumors (NCT05258266).‘

  • This section is edited, and appropriate references are added

Line 353 give the reference (87) to the MODULATE study earlier, e.g., line 346.

  • This has been edited and reference is added.

Lines 353/361 need references if possible ‘TTI-101 … mCRC (NCT02983578).

  • This has been edited and reference is added.

Lines 372/376 need references ‘A current clinical … NL cell.’

  • This has been edited and reference is added.

Section 10.1 is well referenced, but the first paragraph is fairly old literature (<2009).

  • We are unclear about the question over here?

Lines 442/445 need references ‘A current trial … and surgery (NCT03516708)’.

  • This has been edited and reference is added.

Lines 459/463 need references ‘Another preliminary study … oxidase (NOX2)’.

  • This has been edited and reference is added.

Round 2

Reviewer 1 Report

Comments and Suggestions for Authors

The authors have now addressed all the issues raised and made significant changes in the manuscript. The article is now greatly improved and can be accepted in its current form.

Reviewer 2 Report

Comments and Suggestions for Authors

I'm pleased to let you know that after a careful review, your manuscript is now in excellent condition for publication. All of the earlier concerns have been successfully addressed by the revisions, greatly improving the paper's overall quality and clarity. Your research's breadth and depth are noteworthy for their value and depth, contributing notably to the field.

I appreciate your commitment to polishing the manuscript. I do not doubt that the academic community will value your work and that it will significantly advance current discussions and research in your area.

Reviewer 3 Report

Comments and Suggestions for Authors

Acceptable for publication